# The Adipokine Network in Rheumatic Joint Diseases

**DOI:** 10.3390/ijms20174091

**Published:** 2019-08-22

**Authors:** Mar Carrión, Klaus W. Frommer, Selene Pérez-García, Ulf Müller-Ladner, Rosa P. Gomariz, Elena Neumann

**Affiliations:** 1Department of Cellular Biology, Faculty of Biology, Complutense University, 28040 Madrid, Spain; 2Department of Rheumatology and Clinical Immunology, Justus-Liebig-University Giessen, Campus Kerckhoff, 61231 Bad Nauheim, Germany

**Keywords:** adipokine, rheumatic diseases, inflammation, osteoarthritis, rheumatoid arthritis

## Abstract

Rheumatic diseases encompass a diverse group of chronic disorders that commonly affect musculoskeletal structures. Osteoarthritis (OA) and rheumatoid arthritis (RA) are the two most common, leading to considerable functional limitations and irreversible disability when patients are unsuccessfully treated. Although the specific causes of many rheumatic conditions remain unknown, it is generally accepted that immune mechanisms and/or uncontrolled inflammatory responses are involved in their etiology and symptomatology. In this regard, the bidirectional communication between neuroendocrine and immune system has been demonstrated to provide a homeostatic network that is involved in several pathological conditions. Adipokines represent a wide variety of bioactive, immune and inflammatory mediators mainly released by adipocytes that act as signal molecules in the neuroendocrine-immune interactions. Adipokines can also be synthesized by synoviocytes, osteoclasts, osteoblasts, chondrocytes and inflammatory cells in the joint microenvironment, showing potent modulatory properties on different effector cells in OA and RA pathogenesis. Effects of adiponectin, leptin, resistin and visfatin on local and systemic inflammation are broadly described. However, more recently, other adipokines, such as progranulin, chemerin, lipocalin-2, vaspin, omentin-1 and nesfatin, have been recognized to display immunomodulatory actions in rheumatic diseases. This review highlights the latest relevant findings on the role of the adipokine network in the pathophysiology of OA and RA.

## 1. Introduction

According to the World Health Organization “Musculoskeletal disorders comprise more than 150 diagnoses that affect the locomotor system—that is, muscles, bones, joints and associated tissues such as tendons and ligaments—and are the second largest contributor to disability worldwide [1]. Musculoskeletal conditions generally comprise disorders that affect joints such as osteoarthritis (OA) and rheumatoid arthritis (RA). Although both diseases have an inflammatory component, the underlying pathological mechanisms are different.

OA is the most common joint disease affecting 18% of the population above 60 years of age although young individuals, especially juvenile athletes, can also suffer from this disorder [2]. It has been demonstrated that it is triggered mainly by biomechanical stress and joint overload, although obesity and metabolic disease are also considered key risk factors for its development [3,4,5]. Cartilage, subchondral bone, and synovium are the main tissues involved in OA pathogenesis. In recent years, the important contribution of pro-inflammatory mediators such as cytokines, reactive oxygen species (ROS), nitric oxide, and matrix degrading enzymes have been reported [6].

RA is a severe autoimmune disorder, characterized by chronic inflammation of diarthrodial joints, leading to cartilage and bone destruction. RA affects 1% of the population worldwide and is related to loss of physical function, quality of life and high prevalence of comorbid conditions [7]. RA develops in genetically susceptible individuals, under the influence of environmental factors, as well as with the involvement of epigenetic mechanisms. It is a heterogeneous disorder with diverse pathogenic mechanisms and variable clinical forms [8,9].

Unraveling the mechanisms that underlie both immune regulation and resolution of inflammation is crucial for the design of new strategies to treat these two highly prevalent rheumatic disorders. One of the approaches to understand these mechanisms is the study of the neuro-endocrine-immune systems balance, which is crucial for the maintenance of homeostasis and the effective adaptation to stressors [10,11]. This intricate communication and their regulatory mechanisms rely in the presence of common mediators such as neurotransmitters, cytokines, hormones and their receptors. In this sense, on the one hand, dysregulation in immune-endocrine integrated circuitries have been involved in the development of chronic metabolic diseases, comprising obesity, diabetes, and metabolic syndrome [12]. On the other hand, RA has been defined as an example of a disease resulting from abnormal interactions between these systems [13]. Reversal of abnormal cellular phenotypes in this disease has also been shown: As an example, synovial fibroblasts (SF) from RA patients (RASF) could be reprogrammed from a cartilage-degrading phenotype towards a regulatory type by hormones/neurotransmitters [14].

Interestingly, adipose tissue has been shown to be not only involved in the energetic homeostasis but also to act as an endocrine organ by secreting a diverse array of factors referred as adipocytokines. Products of adipose tissue include adipokines, cytokines, chemokines and complement factors [15]. Adipokines are bioactive proteins, which have emerged as modulators of inflammatory and immune response, exerting key roles in rheumatic diseases, both at local and systemic level. Adipose tissue is not the only source of adipokines. Immune cells, chondrocytes as well as synoviocytes also synthesize these mediators. In musculoskeletal disorders, autocrine, paracrine and endocrine pathways acting on target cells and tissues such as bone, cartilage and synovial membrane have been described [16]. Higher levels of adipokines in both serum and synovial fluid from RA patients compared with healthy controls have been reported [17,18]. Moreover, an inflammatory profile in the infrapatellar fat pad (IPFP) of OA patients has been described [19]. Overall, increasing evidence postulates that adipokines play an important role in both immune-mediated rheumatic disease and in degenerative disorders such as OA. In the next sections, we will disclose the most recent data about the role played by these mediators in these two rheumatic diseases.

## 2. Pathophysiology of Osteoarthritis and Rheumatoid Arthritis

### 2.1. Osteoarthritis

OA is a chronic rheumatic disease, considered one of the leading causes of substantial physical and psychological disability worldwide. OA is a complex multifactorial disorder [20,21], where mechanical, genetic, biological, biochemical and metabolic factors are involved. It is characterized by cell stress and extracellular matrix (ECM) degradation, resulting in an imbalance in joint tissue metabolism [22,23,24]. Although cartilage degradation is the main event in the pathology, OA affects the whole joint, including the remodeling of adjacent subchondral bone, osteophyte formation and synovial inflammation, which might culminate in pain, loss of joint function, and disability in advanced stages [21,24,25,26,27,28,29,30,31].

OA cartilage is characterized by an increase of ECM remodeling, cartilage calcification and angiogenesis [32,33]. Chondrocytes have receptors for responding to mechanical stress and inflammatory mediators, many of which are also receptors for ECM components, including cartilage-degradation products [31,34,35,36]. Activated chondrocytes acquire a hypertrophic phenotype, proliferating and increasing the release of inflammatory cytokines and chemokines, stress-induced intracellular signals, ROS, and matrix-degrading enzymes, including aggrecanases and matrix metalloproteinases (MMPs). In addition, ECM protein production is decreased. Depletion of aggrecan and degradation of type II collagen are the main events in cartilage destruction when the process becomes irreversible [37,38,39,40].

Changes in subchondral bone are related to cartilage remodeling, thus playing an important role in OA progression by the release of catabolic mediators that promote an abnormal metabolism in chondrocytes [30,41]. Production of inflammatory and degradative mediators by joint cells also induces the synovial inflammation. Synovitis is characterized by synovial hyperplasia, with SF and synovial macrophage proliferation, as well as immune cell infiltration. These cells release inflammatory mediators including interleukin-1β (IL-1β) and tumor necrosis factor-α (TNF-α), among other cytokines and chemokines which aggravate inflammation [31,42,43]. In addition, SF produce matrix-degrading enzymes, also contributing to cartilage ECM degradation [28,29,44].

### 2.2. Rheumatoid Arthritis

RA is a severe and chronic systemic inflammatory autoimmune disease with unknown etiology that mainly affects peripheral joints symmetrically, leading to progressive articular damage and joint dysfunction. One of the hallmarks of RA is the persistent inflammatory infiltration of the synovial sublining layer that contributes to generate a microenvironment in which stromal cells display a hyper-activated phenotype, releasing several pro-inflammatory and tissue damaging mediators to the joint space [45,46]. Although many of the underlying causes of RA are still unclear, systemic and local immune dysregulation is considered to orchestrate its pathogenesis. In fact, this disease is characterized by the production of autoantibodies such as anti-citrullinated protein antibody (ACPA) and rheumatoid factor (RF), as well as by increased levels of e.g., TNF-α, IL-1β and IL-6 which are recognised as central pro-inflammatory cytokines involved in chronic synovitis, osteoclast formation and subsequent erosive joint damage. In this context, SF and synovial macrophages are recognized to play a key role in driving RA pathology [8,47,48].

RASF characteristically exhibit an autonomous pathogenic phenotype that includes the capability for hyperproliferation and migration, thus contributing to synovial hyperplasia and spreading RA to unaffected joints [8,47,49]. Likewise, resident and monocyte-derived RA synovial macrophages display a pro-inflammatory profile that has been linked to the pathological activity of RASF, and the number of these cells in the synovium of affected joints has been shown to correlate with disease activity and joint erosion [50,51,52]. Furthermore, it is generally accepted that endothelial cells under synovial inflammatory conditions also contribute to chronic synovitis via angiogenesis and recruitment of immune cells in RA [53].

## 3. Adiponectin

Adiponectin has previously been described as an anti-inflammatory adipokine mainly produced and secreted by adipocytes [54]. The highly complex adiponectin molecules exist in different isoforms, the globular form, the trimer (low molecular weight, LMW), the hexamer (middle molecular weight, MMW) and the multimeric (high molecular weight, HMW) adiponectin. LMW, MMW and HMW represent the main circulating forms of adiponectin while the monomer only seems to occur as an intermediate in adipocytes [55].

Two receptors are mainly responsible for adiponectin signaling and the adiponectin isoforms differ in their affinity to the respective receptors [56]. Other receptors such as T cadherin [57,58,59,60] and PAQR3 [61] have also been discussed. Mainly anti-inflammatory effects at the systemic level have been described for adiponectin in atherosclerosis but also for example in metabolic syndrome, type 2 diabetes mellitus [56,62]. In contrast, adiponectin seems to have opposing effects on effector cells of arthritis. Adiponectin was one of the first adipokines to be evaluated in the pathophysiology of arthritis. Numerous studies found that cultured RASF respond to adiponectin by an increase of pro-inflammatory factors including prostaglandin E2, IL-6, IL-8, MMPs-1, -13 [56,62]. However, a stronger pro-inflammatory effect of the HMW isoforms has been described in contrast to LMW adiponectin in these cells [56,62]. For some cell types, even opposing effects have been described for specific adiponectin isoforms. For example, IL-6 secretion was induced by HMW adiponectin in human monocytes (but had no such effect on lipopolysaccharide (LPS)-activated monocytes), while LMW adiponectin reduced IL-6 and increased IL-10 secretion in LPS-activated monocytes [63]. Other cell types including chondrocytes, endothelial cells and lymphocytes showed a mainly pro-inflammatory response to adiponectin [56,62]. Bone cells are affected as well by adiponectin in both OA and RA [15].

### 3.1. Adiponectin in OA

A recent meta-analysis showed that the systemic adiponectin concentration is higher in OA patients compared to healthy controls [64] and a cross-sectional study described that serum adiponectin and leptin were significantly and negatively associated with bone mineral density in OA of the knee in contrast to resistin [65]. Furthermore, serum adiponectin levels were found to be significantly lower in OA patients with metabolic syndrome when compared to OA without metabolic syndrome [66] independent of the body mass index (BMI) [67]. In the context of OA, adiponectin was associated with pain while resistin and visfatin were mainly associated with disability [68]. Interestingly, adiponectin serum levels were shown to be negatively associated with serum MMP-13 in OA-related knee structural abnormalities [69]. The synovial tissue and IPFP of OA patients with metabolic syndrome secreted less adiponectin compared to those OA patients with metabolic syndrome, whereas leptin was increased in OA patients with metabolic syndrome. Adipokine secretion by tissue reflected the systemic adipokine levels observed in these patients [66]. In another study, evaluation of perisynovial and infrapatellar adipose tissue depots revealed differences that were influenced by the BMI of the patient. Compared to adipocytes in the IPFP and synovium of lean OA patients, obese patients showed significantly larger adipocytes with increased synovial fibrosis, macrophage infiltration and toll-like receptor (TLR) 4 gene expression in those patients while adiponectin expression in the synovium was lower in obese patients compared to lean patients [70].

### 3.2. Adiponectin in RA

Systemic adiponectin levels have been described to be increased in chronic-inflammatory RA and to be associated with disease activity and radiographic disease progression [56,62]. However, the correlation with disease activity has been shown in some studies but not by others [62], a discrepancy which still needs to be elucidated. Besides serum levels, synovial fluid concentrations of adiponectin are increased in RA [17,71]. In RA patients, subcutaneous abdominal adipose tissue has been found to secrete more adiponectin compared to subcutaneous abdominal adiposetissue from OA patients, and the amount of adiponectin secreted from this tissue positively correlated with the 28-joint disease activity score (DAS28) and disease duration [72].

Interestingly, RA patients with higher baseline adiponectin showed a more pronounced improvement in inflammatory parameters after anti-TNF-α treatment [73]. Along this line, anti-IL-6 treatment significantly increased adiponectin and reduced chemerin levels in RA patients independently of the disease treatment response [74].

In another recent study, adiponectin indirectly affected T follicular helper cells (Tfh), which did not directly respond to adiponectin, by activating these cells via adiponectin stimulation of RASF, mainly through IL-6. Liu et al. confirm that intraarticular injection of adiponectin increased synovial inflammation with an increased frequency of Tfh in adiponectin-treated collagen-induced arthritis (CIA) mice [75]. Targeting specific adiponectin isoforms using therapeutic antibodies specifically against MMW/HMW adiponectin reduced the IL-6 and IL-8 induction in osteoblasts induced by those isoforms [76]. In addition, Lee et al. showed that antibodies against MMW/HMW as well as against MMW alone significantly ameliorated CIA in mice and that hence both these adiponectin isoforms may contribute to progression of arthritis.

In another recent study, Qian and colleagues suggested a potential cardiovascular protective role of IL-6 inhibition. Adiponectin and osteopontin (OPN) levels were increased and associated with each other in RA serum [77], and RASF stimulation with adiponectin led to a dose-dependent increase in OPN, which in turn caused increased monocyte (RAW264.7) migration. This increased migration could be inhibited by blocking OPN. In vivo, OPN silencing reduced the amount of tartrate-resistant acid phosphatase positive osteoclasts and bone erosion in collagen-induced arthritis (CIA) mice. Therefore, enhanced bone erosion in RA is suggested to be mediated by the induction of OPN in SF, thus increasing the differentiation and recruitment of osteoclast precursors [77].

## 4. Leptin

Discovered in 1994 by Jeffrey Friedman et al. [78], leptin is the main adipokine secreted by adipocytes, with a positive correlation with white adipose tissue mass [79,80]. It is a 16kDa non-glycosylated protein encoded by the *LEP* (*ob*) gene, involved in appetite and obesity regulation by the induction of anorexigenic factors and suppression of orexigenic neuropeptides. Leptin is also implicated in basal metabolism, insulin secretion, bone mass, and reproduction among other functions [78,80,81]. In addition to food intake- and eating-related hormones, leptin synthesis is regulated by energy status, sex hormones and inflammatory mediators [82]. Leptin is mainly secreted by adipose tissue but it is able to act peripherally and centrally, in the hypothalamus, by its release to circulation [83]. Its biological effects are mediated by binding to the long form of leptin receptor (LEPR), which belongs to the class I cytokine receptor superfamily [84].

Leptin is considered a pro-inflammatory adipokine involved in the “low-grade inflammatory state” described in overweight and obese people [85]. In the immune system, leptin modulates both innate and adaptive immunity: it activates proliferation and phagocytosis of monocytes and macrophages, regulates cytotoxicity of natural killer (NK) cells, modulates neutrophils chemotaxis, induces proliferation and inhibits memory T CD4 cells, suppresses type 2 T helper (Th2) phenotype in favour of Th1, and modulates T regulatory (Treg) activity [80,86]. Most immune cells express LEPR at their surface, which has also been described for chondrocytes, SF and osteoblasts [87,88]. In addition, pro-inflammatory cytokines induce leptin synthesis in acute infection and sepsis [89]. Involvement of leptin has been described in several physiological and pathophysiological conditions, including vascular function, reproduction, immunity, and inflammation, as well as in rheumatic diseases [80,90,91,92,93,94].

### 4.1. Leptin in OA

Increased leptin levels in serum and synovial fluid from OA patients have been described to be involved in the physiopathology [95,96,97,98,99] and to be associated with pain and disease severity [99,100]. Single nucleotide polymorphisms (SNP) in the leptin gene and its receptor are linked to OA development [101,102]. Moreover, DNA methylation in OA chondrocytes correlates with leptin expression [103], which associates with the BMI [96]. Fan et al. reported different genes associated with the leptin-induced OA phenotype in rats by microarray analysis, including genes related to MMPs, inflammatory factors, growth factors, apoptosis, and osteogenesis [104].

In OA chondrocytes, leptin increased the production of the pro-inflammatory cytokine IL-1β, as well as matrix-degrading enzymes, including MMP-1, -3, -9, and -13 [80,96,103], suggesting a role in the inflammatory and degradative process that takes place during OA. Accordingly, leptin also increased MMP-2, MMP-9, a disintegrin and metalloproteinase with thrombospondin motifs (ADAMTS)-4 and ADAMTS-5, while it decreased fibroblast growth factor (FGF) and proteoglycan synthesis in rats cartilage [105]. Moreover, leptin induced activation of type 2 nitric oxide synthase (NOS2) in human and mouse chondrocytes, with the involvement of JAK2, PI3K, MEK1 and p38 MAPK signaling [106,107]. By contrast, Dumond et al. showed an anabolic effect of leptin in rat chondrocytes [108]. Leptin also induced changes in chondrogenic progenitor cells causing senescence by the inhibition of their migratory and chondrogenic potential, and the induction of their osteogenic transformation [109].

A role of leptin in OA subchondral bone has also been described, showing an increased expression which is related to high levels of alkaline phosphatase, osteocalcin, collagen type I and transforming growth factor β (TGF-β) [110]. Finally, in relation to the inflammatory process, an increase of IL-6, IL-8 and chemokine ligand 3 (CCL3) by leptin has been described in CD4^+^T cells from OA patients [111]. In addition, Griffin et al. showed that leptin-impaired mice did not develop knee OA inflammation [112].

### 4.2. Leptin in RA

In RA, higher leptin serum levels are related to disease course and activity [113,114,115,116]. In addition to the activity, leptin in serum and synovial fluid has also been linked to disease duration and joint erosion [117].

Several authors have reported the involvement of leptin in joint inflammation by the modulation of different inflammatory mediators. Correlation of leptin and IL-17 has been reported in plasma from RA patients [118]. In addition, leptin induced IL-6 and IL-8 expression in SF from RA and OA patients, where JAK2/STAT3, NF κB, and AP-1 signaling pathways were involved [119,120]. Leptin also increased expression of the vascular cell adhesion molecule (VCAM)-1 in human and murine chondrocytes, and is involved in leukocyte extravasation during RA and OA inflammatory processes. JAK2, PI3K and MAPK intracellular signaling was shown to play a role in this context [87]. Moreover, induction of MAPK signaling by leptin has also been described in RASF [120].

Correlation of leptin with RA pathology has been shown in animal models. Busso et al. reported less severe arthritis in leptin-deficient mice, accompanied with a reduction in IL-1β and TNF-α levels [121]. In addition, leptin injection in CIA mice increased Th17 response, exacerbating RA severity and increasing synovial hyperplasia and joint damage [122]. Serum leptin concentrations also correlated with body fat percentage in RA patients, working as an obesity marker [123]. Cardiovascular risk is another factor associated to obesity and RA pathology. Accordingly, Batun et al. reported an association between leptin and IL-6 concentrations with cardiovascular risk in these patients [124].

## 5. Resistin

Resistin is a dimeric cysteine-rich protein that circulates as a 108-amino acid homodimer in human blood [125]. In humans, this adipokine is mainly produced by macrophages whereas in mice it is primarily secreted by adipocytes [126]. In line with this, human resistin has been primarily associated with inflammatory responses by promoting immune cell recruitment [127,128,129], whereas in mice it has been mostly linked to the development of type 2 diabetes and obesity-mediated insulin resistance [130,131]. However, conflicting results have been published regarding the modulatory role of resistin on inflammatory responses, and whether this adipokine induces anti- or pro-inflammatory effects on cells may depend on both the tissue and organ context and the disease studied [125]. Nevertheless, it is accepted that resistin is generally involved in inflammation and insulin resistance, and subsequently in the development of different pathologies such as coronary artery disease, atherosclerosis, type 2 diabetes, psoriasis and colorectal cancer [132,133,134,135,136]. In this sense, resistin has also emerged as an adipokine implicated in the pathogenesis of OA and RA given its immunomodulatory effects and the ability to enhance the activated phenotype of the effector cells involved in these rheumatic diseases [98,129].

### 5.1. Resistin in OA

Numerous clinical studies have revealed that serum, plasma and synovial fluid resistin levels are increased in OA patients compared with healthy subjects [98,137], suggesting that this adipokine may act as a linker between the inflammatory process and the altered metabolism of joint tissues in the pathogenesis of OA [138].

Resistin levels in the synovial fluid of OA joints have been found to exhibit a positive association with symptomatic and radiographic severity as well as with articular cartilage damage. Moreover, such levels of resistin have been reported to correlate with resistin released from cultured OA cartilage [139,140,141]. Likewise, presence of inflammatory and catabolic factors in synovial fluid, including IL-6, MMP-1, MMP-3 and collagen type II C-telopeptide fragments, also exhibited a positive correlation with synovial fluid resistin levels [141,142]. Furthermore, these adipokine levels have been linked with pain and disability in OA patients with join effusion [143], although no association with knee cartilage volume has been found [97,144].

Interestingly, in vitro studies have shown that resistin-stimulated human articular chondrocytes display an upregulated expression of several pro-inflammatory mediators, including TNF-α, IL-6 and IL-12 [145], and different cartilage catabolic enzymes and mediators, such as MMP1, MMP3, ADAMTS-4 and inducible cyclooxygenase (COX)-2, as well as a decreased production of some components of cartilage ECM such as type II collagen and the proteoglycan aggrecan [146,147,148]. This ability of resistin to promote catabolic over anabolic activity in OA chondrocytes has been confirmed in a recent study, which further found that resistin modulates the expression of several microRNAs (miRNA) involved in the pathogenesis of OA [149]. Accordingly, resistin stimulation of meniscal tissue explants from OA patients resulted in a significant increase of sulfated glycosaminoglycan depletion [150] similar to the dose-dependent loss of proteoglycan in murine cartilage [148]. Furthermore, resistin has also has been found to induce the expression of monocyte chemoattractant protein-1 (MCP-1) by SF from OA patients and to subsequently promote the monocyte migration and infiltration in synovium [151].

However, some inconsistent results have been published regarding the involvement of resistin circulating levels in the pathophysiology of OA. On the one hand, serum levels of resistin have demonstrated a positive correlation with bone marrow lesions and cartilage degradation, also showing an association with different scoring systems for measuring severity, progression and pain in OA [141,152]. Likewise, plasma resistin levels in OA were shown to be associated with progression of radiographic knee [153]. On the other hand, other authors have not found any significant relationship between serum resistin and OA radiographic severity, bone mineral density, pain or cartilage damage [65,97,98,142,154]. Nevertheless, such serum levels exhibited a weak but positive association with histological signs of synovial inflammation in OA patients [98]. Therefore, while the potential pathogenic role of serum/plasma resistin levels needs to be further investigated, there is evidence of the involvement of synovial fluid resistin in OA pathogenic mechanisms by promoting pro-inflammatory responses and cartilage catabolic activity.

### 5.2. Resistin in RA

Resistin is present in blood plasma or serum, synovial fluid, and synovial tissue of RA patients. Synovial stromal cells, including SF, and infiltrating immune cells, such as macrophages and B cells, express resistin in joints affected by RA [155,156]. Higher resistin levels in synovial sublining layers and also in synovial fluid from RA patients compared with OA have been described [17,155,157]. Regarding the potential pathological impact of synovial resistin levels in RA, it has been shown that increased levels correlate with RA disease activity, joint damage [71], and inflammation intensity defined by the intra-articular leukocytes count and IL-6 levels [128]. Conversely, in another study, synovial fluid resistin did not show any significant relationship with inflammation degree based on C-reactive protein (CRP) levels [155].

Besides, the available data concerning the circulating resistin levels in RA patients have also generated conflicting conclusions. Although most studies describe no differences between serum or plasma resistin levels in RA patients and controls [71,113,128,156,158], other authors have reported higher resistin serum levels in RA when compared with healthy controls or with OA patients [155,159]. There is evidence that resistin serum levels in RA patients are positively correlated with inflammatory markers, such as erythrocyte sedimentation rate (ESR) and CRP levels, as well as with the degree of disease activity measured by DAS28 [113,155,160]. Another study also demonstrated a positive correlation between circulating levels of this adipokine and TNF-α levels in blood from RA patients [161], whereas no relationship with TNF-α, CRP levels, leukocytes counts, or with other pro-inflammatory cytokines such as IL6, IL8, or MCP-1 was found by other authors [128,155].

Despite these controversial results regarding correlations between resistin levels in circulating blood or in the joint with parameters of disease activity, there is a general agreement that this adipokine is involved in the pathogenesis of RA. In fact, the intra-articular injection of recombinant resistin in healthy mice induced leukocyte infiltration and hyperplasia of the synovia, leading to a joint inflammation similar to human arthritis [128]. Furthermore, in vitro stimulation of human synovial fluid leukocytes and peripheral blood mononuclear cells (PBMC) with resistin induced the secretion of IL-6, IL-1 and TNF-α by an NF κB-dependent pathway, providing a positive feedback circuit in PBMC by stimulating also its own production [128]. Likewise, stimulatory effect of this adipokine on IL-12 and TNF-α release by both murine and human macrophages was demonstrated to be mediated by the NF κB pathway [127]. More recently, the ability of resistin to increase the production of pro-inflammatory chemokines by SF has been shown [162]. Moreover, resistin has been considered as a key factor triggering angiogenesis in RA affected joints through the upregulation of vascular endothelial growth factor (VEGF) expression in endothelial progenitor cells and causing the homing of these cells to the synovium [163].

The potential involvement of resistin in the RA inflammatory cascade is also sustained by the rapid reduction of its serum levels observed in patients after anti-TNF-α therapy, showing a close association with the inflammation marker CRP [164,165]. Accordingly, a downregulation of resistin gene expression in CD4 Th lymphocytes and CD14 monocytes in RA patients responding to TNF-α inhibitor therapy has recently been demonstrated, showing an increased expression in patients who failed to respond to the therapy [166]. In addition, recent studies in a Chinese population have demonstrated the association of SNP in the resistin gene with RA susceptibility as well as with its clinicopathological characteristics [167,168].

## 6. Visfatin

Visfatin is also called pre-B-cell colony-enhancing factor (PBEF) for its ability to promote B cell precursor differentiation (in synergy with IL-7) or nicotinamide phosphoribosyl-transferase (Nampt) due to its enzymatic activity. However, whether altered systemic or local visfatin levels are associated with changes in the nicotinamide adenine dinucleotide content due to the Nampt activity of visfatin is not well studied. It is produced by adipose tissue but also other tissues such as the liver, bone marrow, and muscle tissue. Visfatin can be induced by inflammatory factors such as TNF-α, IL-1β, IL-6, LPS and chemokines as well as itself [15,56,62].

Hypoxia as found in inflamed joints may also play a role since hypoxia-inducible factor 2alpha (HIF-2α) directly induced visfatin in chondrocytes [169]. In turn, visfatin is able to induce a pro-inflammatory response in a large number of different cell types [115]. In contrast to adiponectin, visfatin is primarily pro-inflammatory. These pro-inflammatory responses have also been described in the effector cells of joints affected by arthritis, for example in SF, lymphocytes, monocytes, chondrocytes or bone cells [15,170]. However, although visfatin is increased in both OA and RA, compared to healthy controls, the levels differ between these two diseases. Likewise, the potential to respond to visfatin is similar in OA- and RA-derived cells but the responses, although pro-inflammatory in both cases, differ in strength between RA and OA [11,29].

Visfatin is also associated with insulin-like effects: It regulates insulin secretion, insulin receptor phosphorylation and insulin-related intracellular signaling [171,172] but the work by Fukuhara et al. (2007) that originally reported visfatin to interact with the insulin receptor was retracted. Therefore, as of now, visfatin does not have a known receptor and several studies showed that its effects are at least in part due to its Nampt activity [173].

Another mechanism includes its influence on insulin-like growth factor-1 (IGF-1) function. Amongst others, IGF-1 is involved in cartilage synthesis and repair by stimulating proteoglycan and collagen type II synthesis. Visfatin has been shown to inhibit these IGF-1 functions, independently of IGF-1 receptor activation [174]. Altered levels of miRNAs may be another means how visfatin mediates its effects as a range of miRNAs were either increased (miR-155, -34a, -181a) or decreased (miR-140, -146a) in OA chondrocytes by visfatin [149].

### 6.1. Visfatin in OA

Systemic visfatin levels are increased in OA patients compared to healthy controls [175,176] but serum levels are lower compared to chronic-inflammatory diseases such as RA. However, OASF respond to visfatin even at low concentrations with increased secretion of pro-inflammatory factors such as IL-8, MCP-1 and other chemokines [170]. Visfatin also induced IL-6 and TNF-α in SF from OA patients, which was mediated by the repression of the miRNA miR-199a-5p via different signaling pathways including ERK, p38, and JNK [177].

A cell culture study on OA suggests a catabolic effect of visfatin because treatment of human chondrocytes with IL-1β resulted in an increased synthesis of the matrix-degrading enzymes MMP-3, MMP-13, ADAMTS-4 and ADAMTS-5 and a decreased synthesis of the extracellular matrix component aggrecan [178]. In line with these effects, visfatin significantly reduced viability, induced apoptosis as well as MMP-1 and MMP-13 secretion in OA chondrocytes [149].

### 6.2. Visfatin in RA

In RA patients, systemic visfatin levels have been shown to be increased in comparison to OA patients and healthy donors and a positive correlation between visfatin and RA disease activity and inflammatory parameters such as CRP has been described [56,62,115]. According to a recent study, visfatin expression is associated with reduced atherosclerotic risk in RA patients [179].

Animal models have also shown a potential role of visfatin in RA. In a murine CIA model, visfatin-deficient mice displayed reduced bone destruction, inflammation and disease progression [180]. In this study from Li and coworkers, visfatin has been shown to be required for osteoclastogenesis. In another study, use of the selective inhibitor APO866, which inhibits the Nampt activity of visfatin, reduced the severity of arthritis in a CIA mouse model and the production of pro-inflammatory cytokines in the affected mouse joints [173].

## 7. Other Adipokines in OA and RA

### 7.1. Progranulin

Human progranulin (PGRN, also known as granulin/epithelin precursor) is a glycoprotein of approximately 75–80 kDa which is composed of seven granulin/epithelin repeats (granulins) that can undergo enzymic proteolysis into small homologous subunits. Both full-length protein and its constituent granulin peptides are biologically active although often with anti- and pro-inflammatory actions, respectively [181]. PGRN was originally described as an autocrine growth factor that stimulates chondrocyte differentiation and proliferation [182], as well as endochondral ossification [183,184]. Moreover, PGRN has also been identified as an adipokine with anti-inflammatory properties mainly mediated by its competitive binding to TNF-α receptors (TNFR1 and TNFR2), which disturbs TNF-α-induced responses [80,185].

Multiple studies have reported significantly increased PGRN levels in cartilage, synovial fluid, as well as in serum from OA and RA patients compared with healthy donors, with higher levels in RA [186,187,188]. Regarding the potential pathogenic role of PGRN, a correlation between circulating PGRN levels and disease activity has been shown in RA patients [189]. In this sense, a recent study in Hispanic RA patients found a correlation between changes in serum PGRN levels and RA progression scores over time, although serum concentrations of PGRN did not predict the clinical response to TNF-α-antagonist therapy [190]. The presence of antibodies against PGRN (PGRN-abs) has been detected in sera from patients with different rheumatic diseases, including RA, showing neutralizing effects on PGRN plasma levels [191]. More recently, it has been proved that PGRN-abs positive RA patients exhibit higher disease activity compared to negative patients, and that a pattern of increased rates of PGRN-abs is observed in the serum of RA patients with poorer outcome [192].

PGRN is secreted by a broad spectrum of cells, including adipocytes, macrophages and chondrocytes [193], and an increased expression of this adipokine has been found during chondrocytes differentiation in vitro, in the IPFP from OA patients [186], as well as in cells infiltrating the sublining layer of RA synovium [187]. Hence, the increased expression of PGRN at local sites of inflammation is suggested to be linked to its ability to initiate immune activation by recruiting fibroblasts, macrophages and neutrophils at the site of inflammation [194]. However, on the other hand, it is now generally accepted that PGRN is also an important mediator in the maintenance of cartilage integrity. This adipokine is able to inhibit the ADAMTS-7/-12 mediated degradation of cartilage oligomeric matrix protein by interfering with direct interactions between these catabolic enzymes and their substrate, as well as by inhibiting the TNF-α-induced expression of both ADAMTS in human chondrocytes [195]. Furthermore, treatment with PGRN has been demonstrated to inhibit proteoglycan loss and the expression of catabolic inflammatory biomarkers induced by TNF-α in cultured human cartilage [196]. In fact, PGRN has been described to trigger anabolic pathways in human cartilage and primary chondrocytes by binding to TNFR2, and to inhibit the IL-1β and LPS induced catabolic metabolism in chondrocytes by blocking TNFR1 [193,197]. In addition, a negative modulation of Wnt/catenin signaling by PGRN has been shown, with the consequent reduction of osteophyte formation and cartilage degeneration [198,199]. PGRN has also demonstrated a protective effect on osteoblast differentiation under an inflammatory milieu by blocking the inhibitory effects of TNF-α on this process [200]. Despite its chondro- and osteoprotective potential, increased PGRN levels as observed in both OA and RA patients are obviously not sufficient to compensate for the catabolic effect of other mediators [181].

Interestingly, an association between serum PGRN levels, functional impairment and disease activity has been found in RA patients [189], in which the balance between PGRN and TNF-α also showed a direct correlation with disease progression [186]. In this regard, different animal models of both OA and RA have demonstrated that loss of PGRN expression results in hyper-susceptibility to develop more severe disease phenotypes [185,196]. Along this line, a recent study has showed that the miR-29b-3p promotes disease development and chondrocyte apoptosis in an OA rat model by modulating PGRN expression [201]. Accordingly, administration of recombinant PGRN or the PGRN-derived atsttrin in OA and RA animal models protected against the development of such rheumatic disorders by, at least in part, inhibiting the TNF-α/TNFR signaling in vivo [183,185,196,202,203,204]. Moreover, atsttrin-transduced mesenchymal stem cells have demonstrated to inhibit cartilage degeneration in an OA murine model after intra-articular injection, and to reduce the TNF-α induced expression of pro-inflammatory molecules by human primary chondrocytes [197].

Other studies in animal arthritis models have shown that PGRN also exerts its immunosuppressive effect by promoting the differentiation, proliferation and recruitment of Treg cells under inflammatory conditions, as well as by inducing IL-10 production [185,205,206]. In this regard, more recently, a study described higher levels of PGRN and human B regulatory cells in RA patients, but without finding a correlation between them [189].

### 7.2. Chemerin

Chemerin is expressed as a 163 amino acid residues-long adipokine that becomes activated after hydrolization by cysteine or serine proteases [207]. The precursor is composed of a hydrophobic signal peptide sequence, a cystatin fold-containing domain, and a labile C terminus. Removal of the signal sequence results in a 143-amino acid secreted preform (prochemerin or chem163S) with low biological activity. Different cleaved isoforms of chemerin have been described depending on the location. Cleavage of the last four amino acids from chem163S rise to chem158K, and removal of the C-terminal lysine from chem158K, results in chem157S, the most active chemerin form. In addition, chem156F, a chemerin C-terminal peptide, is functionally active in vitro [208,209]. Anti-inflammatory properties have been described by mouse chem156S, homologous to human chem157S, on macrophage [210] and in a LPS-induced acute lung injury model [211]. Chemerin plays an important role in the development of coronary atherosclerosis, metabolic syndrome and other diseases [212]. It is also involved in innate and adaptative immunity working as a chemoattractant for NK cells, macrophages and dendritic cells [213,214].

In relation to rheumatic diseases, chondrocytes and SF from RA and OA patients express both chemerin and its receptor chemokine-like receptor 1 (CMKLR1) [208,215,216]. In addition, Zhao et al. described the presence of chem156F in synovial fluid samples from patients with OA and RA [208]. Chemerin induced expression of inflammatory and degradative mediators in these cells, including IL-6, CCL2, and MMP-3 in RASF, CCL2, and TLR4 in SF from RA and OA patients, and IL-1β in chondrocytes. In addition, chemerin stimulated SF motility and leukocyte migration to the joint [209,216,217,218].

Cleaved isoforms of chemerin have been described in synovial fluid samples from RA and OA patients [208]. Ma et al. reported higher levels of chemerin in the synovial fluid and synovial membrane of knee OA patients compared to controls. Chemerin levels also correlated with serum levels of CRP and OA severity [219,220]. Similar results were obtained in patients with OA of the temporomandibular joint, with higher levels this adipokine in synovial fluid and correlation with OA severity and pain [221]. A recent study in a rat OA model showed that chemerin aggravated the disease by inducing activation of Akt/ERK signaling, and by increasing the expression of MMP-1, MMP-3, and MMP-13 in IL-1β activated chondrocytes as well as by decreasing their proliferative capability [222].

In RA, chemerin plasma levels correlated with disease activity and BMI, which is a risk factor in the pathology, arising as a biomarker of meta-inflammation [223]. In addition, the IL-6 inhibitor tocilizumab has an anti-inflammatory and antithrombotic/fibrinolytic role and is able to decrease serum chemerin levels in RA patients [74].

### 7.3. Lipocalin-2

LCN2 is an adipokine produced in joint tissues in response to both mechanical loading and inflammatory mediators [218,224,225]. Specifically, LCN2 expression in chondrocytes is induced by IL-1β, LPS, dexamethasone, adipokines (leptin and adiponectin) [218,226,227] and by osteoblast conditioned medium [228], establishing also a feedback regulatory loop with the catabolic factor nitric oxide [229]. In osteoblasts, the inflammatory molecules TNF-α and IL-17 [230] can induce an upregulated expression of this adipokine, which is able to shift the balance between pro- and anti-osteoclastogenic factors toward a more catabolic metabolism [227,231]. In this sense, a recent study confirmed the catabolic effects of LCN2 in osteoblasts and chondrocytes from OA osteochondral junctions, also showing that osteoblasts induced its expression in a paracrine manner [228].

Although there are data pointing to the involvement of LCN2 in the joint pathophysiology of OA and RA, further studies are needed to elucidate its role in human development of such rheumatic diseases. LCN2 concentration was found to be elevated in synovial fluid of patients with OA and RA, with higher levels in RA [224,225,232]. In OA patients, elevated LCN2 levels in synovial fluid and cartilage have been linked to cartilage matrix destruction given its ability to reduce chondrocyte proliferation and to form a covalent complex with MMP-9 that blocks its auto-degradation [224,225,227]. However, studies in mice have shown that LCN2 overexpression in mouse cartilage is not enough to induce OA pathogenesis, and that its absence has no consequences in the induction of cartilage destruction in *Lcn2*-knockout mice [233]. Regarding RA patients, serum levels of this adipokine have been reported to be an indicator for structural damage in early-stage RA, but not for monitoring disease activity [234]. Likewise, induced LCN2 expression in neutrophils by the granulocyte macrophage colony-stimulating factor has been linked with synovial cell proliferation and inflammatory cell infiltration in RA synovium [225]. Of note, glucocorticoids (normally used to treat OA and RA) have been described to induce the expression of LCN2 by mouse chondrocytes in synergy with IL-1, suggesting that this adipokine may mediate some of the degradative effects on cartilage described after prolonged treatment with such drugs [235].

### 7.4. Vaspin

The adipokine vaspin belongs to a serine protease inhibitor family known to be associated with insulin resistance as well as metabolic syndrome [236]. Also, a link between vaspin and artherosclerosis and cardiovascular disease has been suggested [237]. Vaspin is expressed in several tissues including subcutaneous adipose tissue, skin, stomach and skeletal muscle [238]. In OA patients undergoing joint surgery, it was detected in cartilage, synovium and osteophytes [239]. In the same study, it was found that vaspin serum levels exceeded synovial fluid levels in paired samples.

Studies investigating potential inflammation-related effects of this adipokine are not always in agreement, which may be due to differences between the studied diseases including RA, OA, PsA, juvenile idiopathic arthritis and ankylosing spondylitis. For example, vaspin has been shown to be involved in skeletal muscle inflammation [238] and its serum levels were associated with inflammation in RA [165,240] and with the development of clinically manifest RA after follow up (in contrast to other adipokines) [241]. In an animal study, Transgenic mice overexpressing vaspin showed specific changes in metabolism- and inflammation-related markers: Glucose tolerance was improved, the mice were resistant to high-fat diet induced obesity and had lower systemic IL-6 levels [242]. Cellular effects for this adipokine have also been observed, mainly in the context of metabolism. For example, vaspin has been described to modulate adipocyte differentiation and glucose homeostasis [243]. In the context of human coronary atheromatous plaques, the pro-inflammatory phenotype of human macrophages was suppressed by vaspin [244].

Systemically, serum levels of vaspin were found to be higher in psoriatic arthritis (PsA) patients compared to healthy controls [245], whereas levels were lower in OA patients compared to healthy controls [239]. In synovial fluid, vaspin levels were significantly higher in RA compared to OA patients with a tendency to correlate with DAS28 in the RA group [246]. However, an association of serum vaspin with the inflammation markers CRP or ESR could not be identified in this study. Interestingly, inhibiting inflammation in RA patients by short-term treatment with high-dose glucocorticoids increased vaspin levels [165], suggesting an association with inflammation albeit probably not in a causal manner. On the other hand, in juvenile idiopathic arthritis, vaspin level did not differ significantly compared to healthy controls [247] and there was no association between disease activity and vaspin serum levels [248]. Interestingly, in patients with ankylosing spondylitis low vaspin levels were related to endothelial dysfunction [249].

However, vaspin also affected bone cells and chondrocytes *in vitro*: Human osteoblasts were protected from apoptosis [250] and osteoclastogenesis was inhibited in the pre-osteoblast cell line MC3T3-E1 [251], the murine macrophage cell line RAW264.7 and bone marrow-derived cells. Furthermore, in the RAW264.7 cells, vaspin reduced the RANKL-induced expression of cathepsin K and MMP-9 [252]. Vaspin was also able to inhibit the IL-1β [253] as well as the leptin [254] induced production of catabolic and pro-inflammatory mediators in murine or rat chondrocytes, respectively. These are potential pathomechanisms by which vaspin might contribute to certain arthritic diseases.

### 7.5. Omentin-1

Omentin was discovered as a secretory glycoprotein binding to galactofuranosyl residues on microorganisms as well as a lactoferrin-binding protein. Its role in omental adipose tissue, hence its name, was described in patients with Crohn’s disease. But omentin is also highly abundant in plasma of healthy donors [255]. Most studies point towards omentin as an anti-inflammatory molecule. For example, this adipokine showed anti-inflammatory and anti-atherogenic properties in obese individuals [256] and a negative association with inflammatory bowel disease and metabolic syndrome has been described [255,257,258].

As far as rheumatic diseases are concerned, the role of omentin is rather inconclusive. In serum, omentin was found to be higher in juvenile idiopathic arthritis (JIA) [247] and PsA [248] patients compared to healthy controls. Furthermore, serum levels were higher in JIA patients with active synovitis in comparison to those without active joints [247]. On the other hand, in synovial fluid, omentin levels were found to be lower in chronic-inflammatory RA compared to OA [246]. For RA, an association of omentin with the inflammation marker CRP at baseline has been reported [241]. The level of omentin-1 in synovial fluid of OA patients negatively correlated with self-reported pain and physical disability (as measured by the WOMAC score) in OA patients [259], intended to reflect symptomatic severity in OA. This is in line with the observed inverse correlation between synovial fluid omentin-1 and radiographic severity as assessed by the Kellgren-Lawrence grading [260].

In synovial tissue, omentin was expressed in the synovial lining layer as well as perivascularly ; however, no difference was detectable between RA and OA tissues [261]. In RA, omentin concentrations were inversely associated with MMP-3 levels and influenced by different factors such as disease activity but showed no association with endothelial activation and atherosclerosis in this study [262].

The knowledge regarding specific effects of omentin on different cell types in synovial tissue and systemic inflammation is very limited. For example, the response of RA and OA SF towards omentin was very low [67], suggesting that other effector cells may respond to omentin. Interestingly, Calvet et al. observed a non-significant trend between adiponectin and omentin levels was observed when other adipokines were included in the multivariate linear statistical model using the partial correlation coefficient (PCC) for interpretation [143].

### 7.6. Nesfatin

Nesfatin-1 (nesfastin) is an N-terminal 82-amino-acid peptide nucleobindin-2-derived adipokine, involved in satiety induction and energy homeostasis. Nesfatin was first described as an anorexigenic molecule secreted by the hypothalamus [263] but it is also secreted by subcutaneous adipose tissue, gastric mucosa, pancreatic cells, and testes [264].

Higher nesfatin levels have been detected in serum from OA patients compared to controls [265,266]. Moreover, nesfatin serum and synovial fluid concentrations have been associated with radiographic severity in OA [266], where its levels also correlated with CRP and IL-18 in serum and synovial fluid, respectively. In addition, nesfatin has been detected in human and murine chondrocytes, inducing pro-inflammatory mediators such as COX-2, IL-8, IL-6, and CCL3 in chondrocytes from OA patients [95]. In contrast, recent studies also described this adipokine as an anti-inflammatory molecule [264,267] able to reduce cardiovascular risk [268,269].

In relation to RA, a study demonstrated a positive association between nesfastin and rheumatoid factor in RA patients. In addition, nesfatin concentration correlated with MMP-2, and reduced atherosclerosis in these patients [179].

Akour et al.also described nesfatin activation by the endocrine growth factor FGF21 [270], which has been described as a new adipokine related to BMI in RA patients [271]. Association between FGF21 and rheumatic diseases would be of interest as FGF21 ameliorates CIA by regulating oxidative stress and inflammatory response [272,273]. In addition, FGF21, also inhibits macrophage-mediated inflammation [274] and down-regulates Th17-IL17 axis in CIA mice [275]. In relation to OA, FGF21 concentration in serum and synovial fluid is associated with radiographic knee bone loss [276].

## 8. Concluding Remarks

Common consequences of two of the most prevalent rheumatic diseases, OA and RA, are severe long-term pain and loss of locomotor function, with the subsequent negative impact on health care burden and social system. In order to reduce such socio-economic costs it is necessary to progress in the diagnosis, treatment and in the identification of disease severity biomarkers which requires a more detailed understanding of the diverse factors involved in the initiation and progression of these joint diseases. While the complex etiology of OA and RA are still unclear, and despite the differences between the pathophysiology underlying these joint degenerative diseases, it is now generally accepted that they share similar inflammatory pathways that contribute to synovitis and the loss of balance between cartilage and subchondral bone, leading to the progressive destruction of affected joints [6,8,31,45].

This review summarized the more relevant clinical and experimental lines of evidence for the connection between classic and novel adipokines and the cellular and molecular pathogenic characteristics of two of the most common rheumatic diseases, OA and RA. However, given the pleiotropic action of these molecules and their immunomodulatory effects at both local and systemic levels, the study of their role in the pathogenesis and progression of rheumatic diseases is very complex and has generated conflicting results. In this sense, both pro- and anti-inflammatory effects have been associated with the same adipokine, evidencing that their biological actions depend on the inflammatory context and the disease conditions (Figure 1). Moreover, the correlation between the levels of a particular adipokine in the synovial fluid and the activity/progression of a rheumatic disease may disappear when circulating levels are considered. In fact, when it has been studied whether there is an relationship between synovial and serum levels of adipokines with the development of arthritis in autoantibody-positive individuals at risk of RA, only a significant association between serum levels of vaspin and the disease development was found [241], thus evidencing the complexity of assessing their potential as diagnostic biomarkers.

All in all, it is worthy to note that the contribution of adipokines to the pathogenesis of rheumatic diseases should be understood as cross-talking networks that coordinate with other molecular mediators to orchestrate the activity of effector cells involved in OA and RA. In this regard, activated resident cells in joints affected by these rheumatic diseases are probably the main contributors to the increased levels of adipokines detected in synovium, consequently suggesting that antagonizing local specific adipokines may be a viable option for the development of new therapeutic strategies. However, further research is needed to determine the exact contribution of the adipokine network to rheumatic disorders, which would also help to identify which adipokine could be considered a potential diagnostic and/or prognostic biomarker for OA and RA.

## Figures and Tables

**Figure 1 ijms-20-04091-f001:**
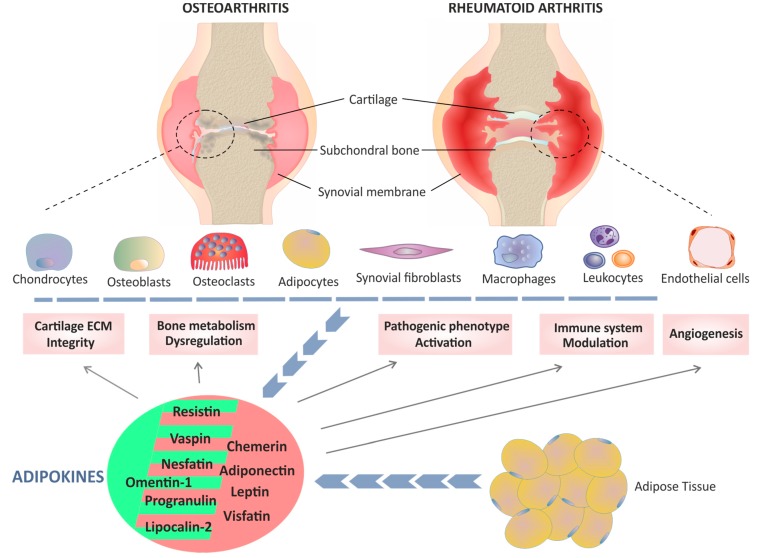
Graphical schematic representation of the role of classic and novel adipokines in two rheumatic diseases: osteoarthritis (OA) and rheumatoid arthritis (RA). A variety of adipokines produced by adipose tissue (blue line of arrows) as well as by chondrocytes, osteoclasts, osteoblasts, synoviocytes and inflammatory cells (blue dashed line and blue line formed by arrows), contribute to multiple pathological mechanisms (solid arrows) involved in the development of OA and RA. Although there are differences between the pathogenesis of OA and RA, common mechanisms have also been identified that affect cartilage extracellular matrix (ECM) integrity, dysregulation of bone metabolism, the pathogenic phenotype of synovial fibroblasts and macrophages, modulation of the immune system and synovial angiogenesis. Depending on the context, both pro- and anti-inflammatory effects have been associated with some adipokines, such as visfatin and resistin. Adipokines with a dominant pro-inflammatory role in arthritis are shown with a light red background, whereas those in which an anti-inflammatory action predominates are shown with a green background.

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
