# Peer review of "The Adipokine Network in Rheumatic Joint Diseases"

_ijms, 2019, doi:10.3390/ijms20174091_

Round 1

Reviewer 1 Report

The white adipose tissue (WAT) is no longer regarded as merely a reservoir for storing excess energy; its secretory activity is currently recognized as being complex and dynamic. A growing body of evidence suggests the involvement of adipokines (adipocytokines) in the pathogenesis of rheumatic diseases. Therefore, the subject of the review is of interest.

The title of the review suggests the analysis of a broader spectrum of rheumatic diseases, whereas the present paper’s main focus is on RA and OA; There is a need for improvement to the English language within your manuscript; Introduction - consider emphasizing the link between body composition and adipokine secretion; Lines 62-64 (pg 2) - the phrase is ambiguous, suggesting that (for ex.) complement factors are adipocytokines. Moreover, adipokines = adipocytokines; Line 64 - bioactiveproteins = bioactive proteins; Reference number 15 does not mention adipokines (!!!); RA pathomechanisms are complex and have been studied extensively. This should be reflected it the section dedicated to RA pathogenesis (which presently contains considerably less information compared to OA in your manuscript); The pathogenic role of certain important cytokines (ex. IL6, TNFα which are mentioned in later sections) is not described; Moreover, important cytokines such as TNFα and IL1 (which play a central role in the pathogenesis of RA) are mentioned for OA (line 100) but not for RA pathomechanism (lines 104-119) in your manuscript (!!!!); "besides" - could be replaced by "moreover", "furthermore", "in addition"; Lines 130, 134 - and others (?) 2. Adiponectin in RA - one third of the section (dedicated to adiponectin) speaks about OPN; the relevance of the adiponectin-OPN relationship in RA pathogenesis is not properly highlighted (consider reorganizing this section with an emphasis on the role of adiponectin); The information in this section is not well organized; throughout the article, different subjects deserve separate paragraphs; Lines 167 - OA tissue (?); Line 172 - "the authors" - consider mentioning the authors along with a brief description of the study; Line 182 - "the group could confirm" (?); Line 186 - "this study" (?); consider mentioning the authors (ex. "Smith et al. showed that..."); Lines 262-263 - "acts as an anti- or pro-inflammatory (mediator)"(?); Lines 309-310 - "in the same study" (?); Visfatin – different subjects deserve separate paragraphs; Line 364 – “insulin effects” (?); Lines 381-384 – “An in vitro study…” – the phrase is ambiguous; Line 512 - "differences between the diseases" - which diseases (?); 4. Vaspin, 7.5 Omentin-1 - consider reorganizing these sections; different subjects deserve separate paragraphs; Line 549 – “active joints” (? – active synovitis?); Lines 552-553 – “synovial fluid of OA patients… disability in OA patients…OA”; Lines 553, 555 – “physical disability”, “radiographic severity” – how were they estimated (interesting subjects which deserve further discussions)? HAQ? WOMAC? Kellgren-Lawrence for Xray changes? Line 556 – “lining layers as well as close to vessels” (perivascular disposition?) – please elaborate; Line 563 – “included in the statistical model” – consider mentioning the authors along with a brief description of the respective study protocol”; Concluding remarks – It would be interesting to summarize the clinical relevance of specific adipokines (either pro- or anti-inflammatory) – can they be considered biomarkers/risk factors/prognostic factors? Is there a specific reason why you chose to describe these 2 rheumatic diseases (but not others)? Did you wish to emphasize certain similarities/discrepancies between a degenerative disease (OA) and an immuno-inflammatory disease (RA)? Are there any differences between seropositive (RF and/or ACPA) and seronegative forms of RA with respect to the involvement of adipokines in the pathogenesis (add figures if possible)?

Author Response

The title of the review suggests the analysis of a broader spectrum of rheumatic diseases, whereas the present paper’s main focus is on RA and OA;

We thank the reviewer for this comment and altered the title accordingly.  

There is a need for improvement to the English language within your manuscript; Introduction - consider emphasizing the link between body composition and adipokine secretion; Lines 62-64 (pg 2) - the phrase is ambiguous, suggesting that (for ex.) complement factors are adipocytokines. Moreover, adipokines = adipocytokines.  Reference number 15 does not mention adipokines (!!!);

We thank the reviewer for this comment and have clarified which mediators are released from the adipose tissue (line 63-64). We agree that often the terms adipokines and adipocytokines are used as synonymes. However, adipocytokines are often defined as the whole secretome of adipocytes which include also cytokines and chemokines. However, according to the suggestion this phrase was now changed.

Reference 15 in the manuscript  is entitled “Adipokines in bone disease”. Nat Rev Rheumatol 2016, 12, (5), 296-302.” by Neumann, E.; Junker, S.; Schett, G.; Frommer, K.; Muller-Ladner, U. This reference summerizes role of adipokines in bone metabolism in different rheumatic diseases.

Line 64 - bioactiveproteins = bioactive proteins;

We apologize for this mistake, we have corrected the spelling mistake.

RA pathomechanisms are complex and have been studied extensively. This should be reflected it the section dedicated to RA pathogenesis (which presently contains considerably less information compared to OA in your manuscript);The pathogenic role of certain important cytokines (ex. IL6, TNFα which are mentioned in later sections) is not described; Moreover, important cytokines such as TNFα and IL1 (which play a central role in the pathogenesis of RA) are mentioned for OA (line 100) but not for RA pathomechanism (lines 104-119) in your manuscript (!!!!);

We appreciate the reviewer’s comment and agree that the information contained in the RA section is not equal to that of the OA. We have modified the text to highlight the unknown etiology of RA and to indicate the important role of the cytokines outlined by the reviewer (Line104; Lines 109-114).

"besides" - could be replaced by "moreover", "furthermore", "in addition";

In accordance with the reviewer's comment, we have replaced “besides” in several areas of the manuscript (line 122, 293, 354, 425, 434, 454).

Lines 130, 134 - and others (?)

By “and others”, we wanted to point out that the listed regulated factors are examples. However, “including” already implies that this list is not all-encompassing. Therefore, it was deleted in line 136 and 140 and replaced with “for example” in line 135.

Adiponectin in RA - one third of the section (dedicated to adiponectin) speaks about OPN; the relevance of the adiponectin-OPN relationship in RA pathogenesis is not properly highlighted (consider reorganizing this section with an emphasis on the role of adiponectin); The information in this section is not well organized; throughout the article, different subjects deserve separate paragraphs;

Our intention was to highlight some recent finding published in the last two years. Due to space limitations, only a selection of recently published data could be included, one of those manuscript evaluating the role of adiponectin with OPN. However, according to the suggestion this area has been reorganized.

 Lines 167 - OA tissue (?);

 “OA tissue” was replaced by “subcutaneous abdominal adipose tissue from OA patients” in order to make it absolutely clear which kind of tissue was meant here (line 173-174).

Line 172 - "the authors" - consider mentioning the authors along with a brief description of the study; Line 182 - "the group could confirm" (?);  Line 186 - "this study" (?); consider mentioning the authors (ex. "Smith et al. showed that...");

We appreciate the reviewer’s suggestions and authors have been mentioned when their studies are described. Additional information was added for the first study mentioned in order to clarify the potential cardiovascular protective effect. As mentioned above, this paragraph was restructured to focus the data more clearly to adiponectin in this section.

Lines 262-263 - "acts as an anti- or pro-inflammatory (mediator)"(?);

Sentence clarified (line 278-279).

Lines 309-310 - "in the same study" (?);

Although the presence of resistin in serum, synovial fluid and synovial tissue of RA patients is described in the same study [155] (“Resistin in rheumatoid arthritis synovial tissue, synovial fluid and serum. Ann Rheum Dis 2007, 66, (4), 458-63.), it is certainly necessary to specify the reference describing the presence of adipokine in plasma. Reference [156] has been specified in the text (line 327).

Visfatin – different subjects deserve separate paragraphs;

Paragraphs were introduced.

Line 364 – “insulin effects” (?);

This was changed to “insulin-like effects” (line 383).

Lines 381-384 – “An in vitro study…” – the phrase is ambiguous;

 “An in vitro study…” was replaced by “A cell culture study...” (line 401)

Line 512 - "differences between the diseases" - which diseases (?);

Information about which diseases we referred to was added (line 547-548).

Vaspin, 7.5 Omentin-1 - consider reorganizing these sections; different subjects deserve separate paragraphs;

Paragraphs were introduced and the structure reorganized as suggested.

Line 549 – “active joints” (? – active synovitis?);

The authors of the publication this sentence refers to do not further specify “active joints”. Therefore, we adopted the term as it is in this context. However, this clinical term is generally used to specify swollen and tender joints. However, the first term was changed as suggested (line 594).

Lines 552-553 – “synovial fluid of OA patients… disability in OA patients…OA”; Lines 553, 555 – “physical disability”, “radiographic severity” – how were they estimated (interesting subjects which deserve further discussions)? HAQ? WOMAC? Kellgren-Lawrence for Xray changes?

Physial disability was estimated by use of the “Western Ontario and McMaster Universities Arthritis Index” (WOMAC) score.

Radiographic severity was assessed by the Kellgren- Lawrence (KL) grading system.

This information was added to the manuscript (line 598-600).

The respective areas in the manuscript were rephrased.

Line 556 – “lining layers as well as close to vessels” (perivascular disposition?) – please elaborate;

The description was changed in the manuscript as follows: “in the synovial lining layer as well as perivascularly” (line 601).

Line 563 – “included in the statistical model” – consider mentioning the authors along with a brief description of the respective study protocol”;

The authors were mentioned and the type of statistical analysis used in the study was specified (line 610-611).

Concluding remarks – It would be interesting to summarize the clinical relevance of specific adipokines (either pro- or anti-inflammatory) – can they be considered biomarkers/risk factors/prognostic factors?

Is there a specific reason why you chose to describe these 2 rheumatic diseases (but not others)? Did you wish to emphasize certain similarities/discrepancies between a degenerative disease (OA) and an immuno-inflammatory disease (RA)?

Are there any differences between seropositive (RF and/or ACPA) and seronegative forms of RA with respect to the involvement of adipokines in the pathogenesis (add figures if possible)?

We appreciate the reviewer’s suggestions regarding the concluding remarks and we have modified  the text accordingly. Additional information concerning OA and RA have been added in order to justify that we focused on both diseases, also highlighting the similarities in their pathological features that allow them to be analyzed in parallel. In addition, some information has been added regarding the potential of adipokines as biomarkers and their relationship with other biomarkers in RA such as autoantibodies, in reference to the study published by Maijer et al [239].

Reviewer 2 Report

Carrion et al have summarized the information on adipokine network in rheumatic diseases. In addition to popular adipokines including adiponectin, leptin, resistin and visfatin, in this review they described the roles of other adipokines such as progranulin, chemerin, lipocalin-2, vaspin, omentin-1 and nesfatin. I hope the following comments will help them to improve the manuscript.

Comments:           

In my PC, downloaded PDF do not contain Fig.1. Ido not have any comment on the figure because I have not seen it. Line192: LEP (ob) gene should be italic. Line233: T CD4 cells is not familiar. Is it CD4+T cells? Line286: What is “catabolic” cyclooxygenase-2? COX2 is sometimes called inducible cyclooxygenase-2. Line291-292: a significant increase of sulfated glycosaminoglycan depletion [150]. Dose this mean a significant decrease of sulfated glycosaminoglycan? Is sulfated glycosaminoglycan in[150] the same as or similar to aggrecan in [146-148]? Visfatin has Nampt activity. Were there any changes in NAD content (concentration) in the systemic blood and in the affected region of RA and OA? Line357: hypoxia-inducible factor2a-alpha should be hypoxia-inducible factor 2alpha. There are granulin molecules derived from progranulin by protease digestion. Is progranulin present in the physiological and inflammatory conditions? Are both progranulin and granulin simultaneously present? Any relation between granulin and rheumatic diseases? Chemerin is the 163 amino acid (AA) protein, and at least two cleaved isoforms of chemerin 157AA and 156 AA are generated, as described in line 466. It is suggested the 157 chemerin shows highest activity, while 156 chemerin is less active. So the authors should distinguish the respective cleaved isoforms of chemerin in their description, if appropriate. Lipocallin-2 is suggested to regulate Iron metabolism and circulating Iron levels are decreased in RA patients. Thus, I wonder is there any relation between Iron metabolism linked to Lipocallin-2 function and RA and OA? What is the degradative effects in [233]? Could you find possible link? Line565: nucleobidin2 should be nucleobindin2, satietyinduction should be satiety induction. Nesfatin-1 is the N-terminal 82 AA of nucleobindin2. Is there any relation between nucleobindin2 as a precursor and RA or OA? FGF21 is an endocrine growth factor that activates Nesfatin neuron. Effects of FGF21 and FGF19 and 23 onRA and OA might be interesting. Abbreviation NK: Natur killer should be Natural killer. COX-2, HIF-2aalpha as aforementioned.

Author Response

Comments: 

In my PC, downloaded PDF do not contain Fig.1. Ido not have any comment on the figure because I have not seen it.

Line192: LEP (ob) gene should be italic.

We have corrected this typographical mistake (line 208).

Line233: T CD4 cells is not familiar. Is it CD4+T cells?

We thank the referee for this comment. In order to avoid similar doubts to those raised in the reviewer, we have replaced T CD4 cells by CD4+T cells (line 249).

Line286: What is “catabolic” cyclooxygenase-2? COX2 is sometimes called inducible cyclooxygenase-2.

We apologize for this mistake, we have replaced “catabolic” by “inducible” (line 302).

Line291-292: a significant increase of sulfated glycosaminoglycan depletion [150]. Dose this mean a significant decrease of sulfated glycosaminoglycan? Is sulfated glycosaminoglycan in[150] the same as or similar to aggrecan in [146-148]?

Regarding the effects of resistin on components of cartilage ECM, this adipokine has been shown to induce substantial sulfated glycosaminoglycans depletion from human meniscal tissue explants [reference 150]. Reference 146 describes that resistin-treated human articular chondrocytes decreased the expression of  aggrecan which is composed by numerous chondroitin sulfate and keratan sulfate chains, and considered the major proteoglycan in the articular cartilage. Regarding results reference 148, authors demonstrated that recombinant resistin inhibited proteoglycan synthesis in human cartilage explants as assessed by [35S]-sulfate incorporation. Therefore, the conclusion extracted from [150] is refered to meniscal sulfated glycosaminoglycan whereas aggrecan or proteoglycan  expression in [146-148] is refered to chondrocytes. This information is specified in the manuscript (“meniscal tissue” and “resistin-stimulated human articular chondrocytes”).

Visfatin has Nampt activity. Were there any changes in NAD content (concentration) in the systemic blood and in the affected region of RA and OA?

The information on the NAD content specifically in affected areas is of interest but not well studied. This was added in line 369-370.

Line357: hypoxia-inducible factor2a-alpha should be hypoxia-inducible factor 2alpha.

We have corrected the spelling mistake (line 374).

There are granulin molecules derived from progranulin by protease digestion. Is progranulin present in the physiological and inflammatory conditions? Are both progranulin and granulin simultaneously present? Any relation between granulin and rheumatic diseases?

In accordance with the reviewer’s questions relative to progranulin/ granulin molecules, we have added additional information about this glycoprotein (lines 419-423).

Chemerin is the 163 amino acid (AA) protein, and at least two cleaved isoforms of chemerin 157AA and 156 AA are generated, as described in line 466. It is suggested the 157 chemerin shows highest activity, while 156 chemerin is less active. So the authors should distinguish the respective cleaved isoforms of chemerin in their description, if appropriate.

Unfortunately, there is very limited information available which distinguish between both isoforms in the context of arthritis. In general, the role of chemerin in the context of RA and OA are mostly limited to measuremet of sytemic levels. Therefore, the requested information is difficult to provide with the exception of the cited study (212). Nevertheless, we have included additional information describing different cleaved isoforms of Chemerin (line 480-488).

Lipocallin-2 is suggested to regulate Iron metabolism and circulating Iron levels are decreased in RA patients. Thus, I wonder is there any relation between Iron metabolism linked to Lipocallin-2 function and RA and OA? What is the degradative effects in [233]? Could you find possible link?

We thank the reviewer for this comment and we find it an interesting aspect, but due to the large amount of information relating to this adipokine, it is necessary to establish a limit on the aspects covered in the text. So we consider that incorporating additional text relating to the role of iron in these diseases would result in an excessive volume of information.

For the same reason, in order not to overload the text with more data, we have not specified the “degradative” effects of glucocorticoids refered in [233] (Reference 235 in the new version of the manuscript). In the “introduction” section of that paper it is written “… side effects caused by GC medication are still common. At cartilage level GC was found to develop different deleterious actions. For instance, it was demonstrated that dexamethasone decreased the viability of ATDC5 mouse chondrocytes [1]. Moreover, dexamethasone also induced the apoptosis of human chondrocytes through the activation of caspases -3, -8 and -9 and the inhibition of Akt-PI3K signaling pathway [2]. Also, GCs impair the proliferation and differentiation of chondrocytes in the growth plate, which might be associated with growth retardation in children treated with these anti-inflammatory drugs [3, 4, 5]…”

Line565: nucleobidin2 should be nucleobindin2, satietyinduction should be satiety induction.

We have corrected both spelling mistakes (lines 613-614).

Nesfatin-1 is the N-terminal 82 AA of nucleobindin2. Is there any relation between nucleobindin2 as a precursor and RA or OA?

According to the reviewer, we have clarified that Nesfatin-1 (nesfastin) is derived from nucleobindin-2 (line 613). We have not found any reference about the relation between nucleobindin2 and RA or OA.

FGF21 is an endocrine growth factor that activates Nesfatin neuron. Effects of FGF21 and FGF19 and 23 onRA and OA might be interesting.

We thank the reviewer for this comment. We have included a new paragraph regarding this issue (line 627-632)

Abbreviation NK: Natur killer should be Natural killer.

COX-2, HIF-2alpha as aforementioned

We have corrected the respective spelling mistakes.